# Reexamining the Environmental Kuznets Curve in Chinese Cities: Does Intergovernmental Competition Matter?

**DOI:** 10.3390/ijerph192214989

**Published:** 2022-11-14

**Authors:** Zhenbo Zhang, Mengfan Yan

**Affiliations:** School of Public Administration, Nanjing Audit University, 86 West Yushan Road, Nanjing 211815, China

**Keywords:** environmental Kuznets curve (EKC), China, intergovernmental competition (IGC), target responsibility system (TRS), official ranking tournament

## Abstract

Since China’s central authority began enforcing the environmental target responsibility system and introduced environmental indicators to the official ranking tournament in 2007, an ecological transformation has emerged in the intergovernmental competition (IGC) among localities. Because the extant literature on the environmental Kuznets curve (EKC) remains unclear regarding how that ecological IGC transformation changes the EKC economy–pollution correlation, this research investigates the degree to which the transformed IGC changes the form of the EKC, and how that altered EKC varies for different pollutants (i.e., SO_2_ and CO_2_) and in different regions (i.e., the eastern, central, and western regions). The results demonstrate a consistently inverted U-shaped relationship between income and SO_2_ emissions in all three regions, whereas when CO_2_ emissions are taken as the pollution indicator, the EKC hypothesis holds only in the eastern and central cities, and a positive linear income–CO_2_ nexus is found in the western region. Spatial analysis reveals that whereas the IGC flattens the inverted U-shaped curves between income and SO_2_ emissions, it has led to a higher economic cost, corresponding to the turning point of the EKC for CO_2_ emissions. The findings indicate that the ecological transformation of the IGC has facilitated a positive up–down yardstick competition in the strategic interactions of sustainable development across local Chinese governments, which can lead to a kind of balance between centralization and decentralization by inspiring local officials’ adaptability and activity in reducing pollutant emissions and strengthening the officials’ responsiveness to performance rankings. This study elucidates the environmental impacts of IGC in China and provides an institutional explanation for the strategic interactions among local governments when they are tackling the environment–economy nexus under multitask conditions.

## 1. Introduction

The environmental Kuznets curve (EKC) hypothesis postulates that environmental degradation increases during the early stage of economic development and then slows with further economic growth. Although a large body of literature has touted the EKC hypothesis and regarded it as an established fact [1], it remains controversial whether the inverted U-shaped income–pollution nexus exists in China, and if it does, when and how the country can reach the threshold of the EKC [2,3,4]. One factor leading to this controversy is an ignorance of the potential impacts of spatial dependence in pollutant emissions among localities––impacts that can arise from the strategic interactions among local governments regarding how to balance the enforcement of top-down environmental policies with traditional promotion-oriented intergovernmental competition (IGC) [5,6,7].

The direct environmental effects of the IGC began in 2007, when China’s central authority enforced the environmental target responsibility system (TRS) and introduced environmental indicators to the official ranking tournament. Under the environmental TRS, the central authority first set assignment indicators and emissions reduction targets for subnational governments, and then it ranked local officials according to their performance in meeting those targets, accordingly attaching certain punishments to the officials that were at the lowest performance ranking [8]. As stated in the subsequently launched veto mechanism, the poorest performers would probably lose their opportunities for promotion in the next political cycle. Consequently, although economic growth is still the most important objective in local administrations [9], local officials also tailor their enforcement of top-down environmental policies to balance and align their environmental performance with that of their peer competitors. By doing so, they neither suffer notably more economic damage than their competitors do when fulfilling environmental targets, nor do they lose out on any potential advancement in the promotional tournament as a result of strict environmental enforcement. With this dynamic, we can assume that the ecological transformation of the IGC can prompt local officials to press ahead with deep cuts in pollutant emissions and, accordingly, to flatten the EKC curve.

Whereas some previous literature explored the impacts of spatial effects on the form of the EKC, most studies focused on the effects of pollution diffusion arising from cross-regional industrial transfer between geographically adjacent provinces [2,10,11,12], and there has been little focus on the impacts of the IGC that generally exist among localities with similar economic levels, whether they are adjacent or not. Therefore, we contribute to the literature by investigating the degree to which the ecologically transformed IGC changes the form of the EKC (i.e., the turning point), and how it varies for different pollutant indicators (i.e., for SO_2_ and CO_2_, noting that the SO_2_ was set as the assessment indicator in the TRS, while the CO_2_ was not) and in different regional divisions (i.e., the eastern, central, and western regions, which differ greatly in their economic levels and officials’ priorities). This study elucidates the environmental impacts of IGC in China and provides an institutional explanation for the strategic interactions among local governments when they are tackling the environment–economy nexus under multitask conditions.

This article is structured as follows. The next section introduces the relevant literature and the theoretical framework for the subsequent analysis. Section 3 outlines the study’s research design, including the development of the estimation model and the definitions of the variables, followed by the data collection. Empirical results and a discussion of the findings are presented in Section 4. Finally, Section 5 presents the conclusions that can be drawn from our research and proposes relevant policy implications.

## 2. Literature Review and Theoretical Development

Throughout the academic community, the environmental Kuznets curve has become a common research model for analyzing the relationship between environmental pollution and economic development [1]. However, the assumption of the validity of the EKC in China regarding the inverted U-shaped income–pollution nexus is controversial. On one hand, abundant evidence has been offered to confirm the EKC hypothesis in China [13], although the evidence is inconsistent regarding whether the threshold of the EKC can be reached in recent years or is still far away [2,3,4]. On the other hand, many scholars have provided opposing insights regarding the EKC hypothesis, as they found that the inverted U shape does not hold when depicting the relationship between income and CO_2_ emissions [14,15], haze pollution [16], and solid waste generation [17]. Furthermore, Du et al. [16], Hao et al. [10], Wang and He [7], and others found that the correlation between income and pollution is more likely to be an (inverted) N shape rather than the inverted U shape.

Several types of factors, which generally have been investigated separately in the previous literature, have been verified to influence the validity of the EKC hypothesis, and they include: (1) pollutant indicators––such as flow pollutants (e.g., SO_2_ and NO_x_), which cause immediate damage and elicit constant pressure on the administrative authorities, and stock pollutants (e.g., CO_2_ and heavy metal wastes), which harm the environment later and long into the future, and which some myopic governments allow to grow with income [18]; (2) regional discrepancies––which are especially salient because economic scale and income accumulation remain significantly unbalanced across the eastern, middle, and western regional divisions in China [2,19]; (3) trade openness and technology aggregation––which imply that increasing scale effects arise from production expansion, that composition effects arise due to the shifting of “dirty” industries, and that technique effects occur because of the advancement of technology and distribution [1,13,19]; (4) the industrial and energy structure––which, because China’s energy consumption structure remains coal-dominated, its rapid industrialization would stimulate rigid demands for energy and present sustained growth of pollutant emissions [3,16]; (5) governments’ environmental regulation and policy preferences––which, in the face of fundamentally conflicting objectives between economic growth and environmental preservation, can cause local governments to tend to strategically implement top-down environmental regulations, consequently leading to varying income–pollution nexuses in different localities [3,20]; and (6) pollution diffusion––which is salient because geographically adjacent regions share many characteristics in terms of factor endowments, development patterns, and institutional environments that may cause them to present an interactive strategy for emissions reduction, as well as an interrelated income–pollution nexus [21,22] (as shown in Table 1).

The IGC framework, which simultaneously incorporates the environmental TRS and the official ranking tournament mechanism, could significantly integrate the factors listed above in the EKC research. On one hand, the TRS is designed by the central authority to allocate and ensure the accountability of local officials in policy implementation and emissions reduction. Under the environmental TRS, the central government established general goals for environment performance indicators and allocates subgoals to local governments at all levels [8]. On the other hand, because of the complexity in assessing environmental improvements, a type of performance-ranking tournament is applied to punish those who present poor environmental performance, and it functions as a complement to the traditional economic-oriented, absolute performance-based promotion tournament mechanism [6]. Consequently, under the ecologically transformed IGC pattern, local officials strategically tailor their enforcement of environmental regulations to balance and align their own regulatory performance with that of other competitors [9], especially in localities with similar economic levels, among which there is fierce competition for political promotion [6,23].

In view of these various factors, a spatial analysis needs to be conducted to investigate how the ecological transformation of the IGC changes local officials’ economy–environment preferences under multitask conditions, and accordingly, how the IGC affects the regional income–pollution correlations in the competitors’ jurisdictions (i.e., the EKC hypothesis). Several questions arise from this issue. (1) Because the IGC occurs mostly among local governments with similar economic levels, the spatial autocorrelation of pollutant emissions could also exist beyond geographic boundaries, rather than only in the geographically adjacent jurisdictions (with a shared border). (2) Because localities’ priorities regarding environmental preservation vary and are often affected either by their endogenous developmental stage or by their neighboring/competing peers’ environmental strategies, different EKCs could appear in the eastern, central, and western regional divisions; meanwhile, that could also present significant spatial dependence of pollutants emissions in each region. (3) The emissions mitigation effects of the IGC may hold only for the pollutants that were included in the environmental TRS (e.g., SO_2_ and chemical oxygen demand [COD]), while other pollutants, such as CO_2_, that were not included during our research period could have attracted less attention from local officials and increased along with their peer competition for economic growth. This implies that compared with those of not considering the spatial interactions caused by the IGC, relatively lower (or higher) income levels could exist that correspond to the threshold of the EKC at which the emission amount of SO_2_ (or of CO_2_) changes from upward to downward when the spatial effects are fully accounted for.

## 3. Research Design

Ever since Ehrlich and Holdren [24,25] originally proposed the environmental impacts of population, affluence, and technology (IPAT) equation in their seminal work, extensive studies have examined the environmental impacts of human activities using the IPAT equation [26,27]. The IPAT equation is expressed as
(1)I=P × A × T

However, this equation has been criticized for not allowing any diagnostic analysis. For example, the IPAT equation is incapable of reflecting discrepancies in the environmental impacts of different factors, in addition to the fact that the correlations between impacting factors and the environment are often nonlinear and disproportionate [26]. To overcome the shortcomings of the IPAT equation, Dietz and Rosa [28] transformed the model into a random form, the STIRPAT model, which is written as
(2)$${I=\alpha \;P}^{\beta_{1}}{\;A}^{\beta_{2}}{\;T}^{\beta_{3}}\;\varepsilon$$
where *α* is the model coefficient, *β*_1_, *β*_2_, and *β*_3_ represent the coefficients for population, affluence, and technology, respectively, and *ε* denotes the random error. In this article, the environmental impact (*I*) is measured by the total amount of pollutant emissions (*PE*), which is separately measured by industrial SO_2_ and CO_2_. Affluence is proxied by real per capita GDP (*PGDP*), and population is measured by population density (*PD*). Technology is proxied by the index of regional innovation and entrepreneurship (*RIE*) developed by the Center for Enterprise Research of Peking University (PKUCER, https://www.cer.pku.edu.cn/, accessed on 11 November 2022), which incorporates five measuring subitems (the number of newly-built enterprises, foreign direct investment, venture capital investment, number of patents granted, and number of trademarks registered) and has been used increasingly widely as a proxy for the regional level of technology advances [29].

If the EKC hypothesis holds for the income–pollution nexus, a positive environmental impact from the IGC will reduce the economic costs corresponding to the peak of pollutant emissions. Therefore, the quadratic form of real per capita GDP (*PGDP*), *PGDP*^2^, is introduced into Equation (2). Furthermore, the level of governmental intervention (*GOV*) is introduced in our analysis. Substituting those proxy variables into Equation (2), and introducing the *GOV*, we obtain
(3)$${\mathrm{PE}=\alpha \;\mathrm{PGDP}}^{\beta_{1}}{\;(\mathrm{PGDP}}^{2}{)}^{\beta_{2}}{\;\mathrm{PD}}^{\beta_{3}}{\;\mathrm{RIE}}^{\beta_{4}}{\;\mathrm{GOV}}^{\beta_{5}}\;\varepsilon$$

Taking the natural logarithm of both sides of the equation and integrating Equation (3) with respect to city (*i*) and year (*t*), we ultimately have
(4)$$\ln PE_{\mathrm{it}}{=\beta }_{1}\ln{\mathrm{PGDP}}_{\mathrm{it}}{+\beta }_{2}{\ln{\mathrm{PGDP}}^{2}}_{\mathrm{it}}{+\beta }_{3}\ln{\mathrm{PD}}_{\mathrm{it}}{+\beta }_{4}\ln{\mathrm{RIE}}_{\mathrm{it}}{+\beta }_{5}\ln{\mathrm{GOV}}_{\mathrm{it}}{+\varepsilon }_{\mathrm{it}}$$
where *β*_1_ through *β*_5_ represent the elastic coefficients, indicating that when other impacting factors remain unchanged, a 1% change in one factor can cause a percentage change in pollutant emission, and where *ε_it_* is the stochastic error term.

If a spatial correlation exists, two types of spatial regression model are used to capture and identify the spatial effects: the spatial lag model (SLM), which contains a spatial lag term of the explained variable in a general panel data regression model, and the spatial error model (SEM), which contains a spatial error term. This study sought to capture the spatial correlations of pollutant emissions among cities. For those cities, latent spatial effects exist not only in emissions indicators, but also in economic and other explanatory variables, and both of those two kinds of spatial effects should be accounted for. Together with the econometric considerations in both the SEM and SLM, LeSage and Pace [30] suggested integrating the two models to form the comprehensive spatial Durbin model (SDM). Applying that approach to this research, the SDM is specified as Equation (5), which is written as
(5)$$\ln PE_{\mathrm{it}}=\rho \overset{n}{\underset{j=1}{\sum }}w_{\mathrm{ij}}\ln PE_{\mathrm{it}}{+\beta }_{1}\ln{\mathrm{PGDP}}_{\mathrm{it}}{+\beta }_{2}\ln{\mathrm{PGDP}}_{\mathrm{it}}^{2}{+\beta }_{3}\ln{\mathrm{PD}}_{\mathrm{it}}{+\beta }_{4}\ln{\mathrm{RIE}}_{\mathrm{it}}{+\beta }_{5}\ln{\mathrm{GOV}}_{\mathrm{it}}\;\;\;{+\theta }_{1}\overset{n}{\underset{j=1}{\sum }}w_{\mathrm{ij}}\ln{\mathrm{PGDP}}_{\mathrm{it}}{+\theta }_{2}\overset{n}{\underset{j=1}{\sum }}w_{\mathrm{ij}}\ln{\mathrm{PGDP}}_{\mathrm{it}}^{2}{+\theta }_{3}\overset{n}{\underset{j=1}{\sum }}w_{\mathrm{ij}}\ln{\mathrm{PD}}_{\mathrm{it}}\;{+\theta }_{4}\overset{n}{\underset{j=1}{\sum }}w_{\mathrm{ij}}\ln{\mathrm{RIE}}_{\mathrm{it}}{+\theta }_{5}\overset{n}{\underset{j=1}{\sum }}w_{\mathrm{ij}}\ln{\mathrm{GOV}}_{\mathrm{it}}{+\mu }_{i}{+\nu }_{t}{+\varepsilon }_{\mathrm{it}}.\;$$

In Equation (5), *ρ* indicates whether and how the amount of pollutant emissions of city *j* is impacted by other cities, *β* denotes the influence of a vector of explanatory variables, and *θ* represents the explanatory influence of other cities under the spatial weights matrix *w_ij_* on the pollutant emissions (ln*PE*) of city *j* in year *t*. The terms *μ_i_* and *ν_t_* denote the individual and time fixed effects, respectively, and *ε_it_* is the error term. It is well known that if *θ* = 0, the SDM is the same as the SLM. If *θ* = –*βρ*, the SDM directly becomes the SEM. According to LeSage and Pace [30], the Wald and likelihood ratio (LR) tests should be conducted to test whether the spatial Durbin model is more suitable for carrying out this empirical study.

We compute the matrix *w_ij_* by referring to the characteristics of the IGC among Chinese municipalities. First, because the city-level officials are assessed and promoted by their immediate superior leaders, the IGC generally occurs among municipal governments belonging to one single province. We therefore set the matrix, *Wpro*, to proxy the *w_ij_*, by which to capture the attributes of political affiliation for each city. In the construction of *Wpro*, *w_ij_* is set on one if cities *i* and *j* belong to the same province, and zero otherwise. Second, according to the economy-based official promotional tournament, the closer the economic levels of local governments are, the stronger the IGC will be among these governments’ executive officials [6], and the more attention they will pay to their competing peers’ performance of emissions reduction [31]. Therefore, we set the matrix, *Weco*, to proxy the *w_ij_*, with which to measure the economic disparity in each pair of cities. In the construction of *Weco*, the matrix *w_ij_* is computed by the inverse of absolute values of the difference between the real per capita GDP of city *i* and that of city *j*. Third, combining the abovementioned arguments, we can conclude that the cities that simultaneously belong to the same province and also have similar economic levels should have fiercer intergovernmental competition and accordingly will present more spatial interactions on pollutant emissions. Therefore, we constructed a nested matrix, *Wpe*, to incorporate the two dimensions of impacts.

The study was conducted using a panel data set from 274 cities in China for the period 2007 to 2017. Mainland China currently has 293 municipal cities, and 19 of them are excluded from the analysis due to data unavailability. The data for city-level CO_2_ emissions were calculated by summing the county-level CO_2_ emissions that were estimated by Chen et al. [32]. In his measurement, a particle swarm optimization-back propagation (PSO-BP) algorithm was employed to unify the scale of DMSP/OLS and NPP/VIIRS satellite imagery. The data for industrial SO_2_ are from the China Environmental Statistical Yearbook (2008–2018), and the data for other variables were all from the China Statistical Yearbook (2008–2018). The definitions and descriptive statistics of variables used in this research are listed in Table 2.

## 4. Empirical Results

### 4.1. Spatial Dependence Tests

We used Global Moran’s *I* values to disclose the degree of autocorrelation for variables throughout the study region [33]. Global Moran’s *I* values fall between −1 and 1; when *I* is greater than 0, positive spatial dependence is present, with higher values of Moran’s *I* corresponding to a stronger positive spatial effect, and vice versa. Table 3 presents our test results of Moran’s *I* for global spatial dependence of the amount of pollutant emissions under the three different spatial matrices (i.e., *Wpro*, *Weco*, and *Wpe*). The Moran’s *I* values are all above 0 and statistically significant at a 90% significance level or higher, thus implying the existence of positive spatial autocorrelations in the amounts of both the industrial SO_2_ and CO_2_ emissions in cities belonging to the same province (weighted by *Wpro*), or with similar economic levels (weighted by *Weco*). The results demonstrate that both the industrial SO_2_ and CO_2_ emission levels for all cities are not completely random but instead are in a positively related state of spatial dependence. This relationship will be biased if the estimation models are constructed without spatial effects [34,35].

Moreover, for the industrial SO_2_ emissions, the level of spatial dependence between cities with similar economic development is relatively low (the values of Moran’s *I* range from 0.033 to 0.154). At the same time, when the province factor is incorporated and hereafter the nested matrix *Wpe* is obtained, the value of Moran’s *I* for each year turns out to be greater than those under the *Wpro* or *Weco*. The results therefore reveal that the spatial dependence of SO_2_ emissions is much more apparent among cities that belong to the same province and are at similar economic levels, thus implying a much stronger IGC among them. The values of Moran’s *I* for CO_2_ emissions are also higher under the nested matrix *Wpe* than those under the other two matrices. We can therefore conclude that both of the spatial autocorrelations separately weighted by *Wpro* and *Weco* should be considered in order to capture the spatial dependence of pollutant emissions. Accordingly, the nested matrix *Wpe* was applied in the subsequent spatial estimations.

### 4.2. Spatial Econometric Estimation Results

For comparison purposes, the panel data models both without and with the spatial dependence of city-level SO_2_ emissions (Case 1) and CO_2_ emissions (Case 2) are estimated, and the results are presented in Table 4. We first report the estimation results when spatial effects are not accounted for. The LM test suggests that either the fixed-effects (FE) or random-effects model is more appropriate than the classical model (not reported) in both cases. Furthermore, the Hausman test suggests that the FE model is better than the random-effects model. Finally, the LR test concludes that the two-way FE model is better than the one-way FE model. As the table shows, in both cases, the coefficient of ln*PGDP* is consistently significantly positive and the coefficient of ln*PGDP*^2^ is significantly negative, thus proffering robust evidence that an inverted U-shaped relationship between economic growth and pollutant emissions does exist. On the basis of the results reported in first column, the logarithm of per capita GDP corresponding to the peak of SO_2_ emissions is 10.4464, indicating that SO_2_ emissions in China have crossed the threshold and proceeded to the downward stage as a whole, which supports the findings of Jiang et al. [11], Zhang et al. [31], and Zhao et al. [12]. However, the ln*PGDP* at the peak of CO_2_ emissions is 11.8304, indicating that carbon emissions in most of cities are still increasing rapidly. Similar evidence can be found from Kang et al. [19], Wang and Ye [15], and Yin et al. [3].

According to Anselin [34] and Pinkse and Slade [35], given the condition that the spatial lagged and error correlations do exist, there will be serious consequences from ignoring these spatial correlations. If spatial lag dependence is ignored, ordinary least squares (OLS) estimators will be biased and inconsistent. If spatial error dependence is ignored, OLS estimators will be unbiased but inefficient, and the standard errors of the estimators will be biased. We report the results from verifying spatial dependence in Table 5. According to Elhorst [36], the significant results of the (robust) LM test response testify to the existence of spatial correlations, so an empirical analysis that does not consider the spatial effects would be subject to bias and hence not reliable [2,31]. The results of the Wald and LR tests are presented in the last two columns, and they reject the hypotheses of *θ* = 0 and *θ* = –*βρ*, respectively, thus indicating that the SDM model is more suitable than either the SLM model or SEM model for this analysis [30].

Estimation results from spatial Durbin models are reported in the last two columns of Table 4. The Hausman and LR tests suggest that the two-way FE SDM is the most appropriate model in both cases. As can be seen in the first row, the estimated coefficients of lagged dependent variables (*Rho*) are all positive at the 1% significant level in both cases, thus revealing that both the industrial SO_2_ and CO_2_ emissions of a certain city are likely to be influenced by other competing cities. The estimated coefficients of ln*PGDP* and its quadratic term ln*PGDP*^2^ are significantly positive and negative, respectively, verifying the validity of the EKC hypothesis. The product terms of the explanatory variables and spatial weights matrix *Wpe* in the SDM reflect how these explanatory variables in other correlated cities affect the pollutants emissions of a certain city. In case 1, the coefficients of *Wpe**ln*PGDP* and *Wpe**ln*PGDP*^2^ are significantly positive and negative, respectively, indicating that a city’s industrial SO_2_ emissions are sensitively correlated to its competing city’s economic levels in an inverted U-shaped way. In contrast, as the results in case 2 show, there appear to be no significant spatial correlations in CO_2_ emissions. We can conclude, therefore, that the impacts of IGC exist only for the industrial SO_2_ emissions, and they cannot be identified in the city-level CO_2_ emissions––which is consistent with the factual background that only the industrial SO_2_ emissions, and not the CO_2_ emissions, have been set as the indicator for assessing the environmental performance of local officials.

Table 4 also reports the estimation coefficients of the control variables. One city’s population density (ln*PD*) is not only negatively correlated with its SO_2_ emissions, but it is also negatively impacted by its competing cities’ sulfur emissions, further supporting the spatial interaction effects on SO_2_ emissions among local governments. In contrast, the CO_2_ emissions have been significantly increasing with rising population density, but present no spillover effects across cities. Although the level of regional innovation and entrepreneurship (ln*RIE*) is found to exert a positive but insignificant impact on pollutant emissions, a significant spillover effect of technology aggregation is found in the reduction of CO_2_ emissions but not in the reduction of SO_2_ emissions. We can infer then that the IGC regarding environmental indicators (e.g., the SO_2_ emissions) would have handicapped the distribution of technologies designed for cleaner production and emissions reduction among local governments, which is consistent with the arguments of Eaton and Kostka [23].

Table 6 reports the estimation results from the SDM models conducted based on the regional divisions (i.e., the eastern, central, and western regions). Because the diagnostic results suggest that the two-way FE SDM is the best fit, we used it to limit the interpretation of coefficient estimates. The results presented in first row show that the indicator *Rho* is positive at the 1% significant level in both cases across all the regions, indicating a pervasive spatial dependence of pollutants emissions across cities that present high degrees of IGC. The estimated coefficients of ln*PGDP* and ln*PGDP*^2^ suggest that the EKC hypothesis is valid only in the eastern and central regions, whereas it does not operate in the western region. The results in the final column even reveal a significant positive correlation between economic growth and CO_2_ emissions. In terms of the spillover effects, the estimated coefficient of *Wpe**ln*PGDP* demonstrates that a western city’s CO_2_ emissions are negatively correlated with its competing city’s economic development. The results jointly reveal that carbon emissions have been increasing precipitously with economic growth in western Chinese regions, and that siphoning effects of economic factors in urban development are also reflected in carbon emissions.

It is essential to use factor analysis to estimate the value change of the economic variable at the EKC turning point, so as to evaluate the effects of IGC on the income–pollution correlations. To this end, using the results of SDMs, we further calculate the direct, indirect, and total marginal effects of different influencing factors on SO_2_ and CO_2_ emissions for each regional division. The coefficients of ln*PGDP* and ln*PGDP*^2^ are reported in Table 7, and the resulting economy–emissions nexuses are plotted in Figure 1. Estimation results without spatial dependence are derived from two-way FE models, whereas estimation results identifying spatial effects are derived from the total effects in two-way FE SDMs. In Figure 1, T1 marks the per capita GDP corresponding to the turning point of EKC with accounting for the spatial effects, and T2 marks the GDP at the turning point without accounting for the spatial effects.

As is shown in the first and fifth columns of Table 7 and panels A and B of Figure 1, when taking all of the cities as the sample, the EKC hypothesis is consistently valid in both cases. The level of per capita GDP corresponding to the peak of CO_2_ emissions is higher than that of SO_2_ emissions, whether or not the spatial effects are considered. Those results demonstrate that compared with the SO_2_ emissions, which have already advanced into the downward stage, much greater economic costs will have to be paid to reach the turning point of the CO_2_ emissions. Furthermore, in Panel A of Figure 1, T1 is lower than T2, revealing that the per capita GDP corresponding to the peak of SO_2_ emissions is estimated to be lower when the spatial effects are fully accounted for than when they are not. In contrast, in Panel B, T1 is greater than T2, which demonstrates that the spatial interaction has a negative effect on the reduction of carbon dioxide emissions. These results indicate that the IGC has significantly reduced the SO_2_ emissions but has promoted the CO_2_ emissions, which could be due to the fact that during the research period the reduction of SO_2_ emissions had been set as a veto assessment indicator in the official promotion tournament, while the reduction of CO_2_ emissions had not.

We also report the estimated results for different regional divisions in Table 7, and we illustrate them in panels C–H in Figure 1. Because the SO_2_ emissions reduction was set as assessment indicator in 2007, the municipal governments in all three regional divisions gradually formulated positive IGCs that target the mutual performance of SO_2_ emissions reduction and lead to a lower per capita GDP corresponding to the threshold of EKC than that when the spatial effects are not accounted for. Those results indicate that by assessing official performance with environmental indicators, for SO_2_ emissions, we can achieve the turning point of EKC at a relatively low economic cost––a cost that at least is not as high as that in the eastern cities. In contrast, although the EKC hypothesis holds for CO_2_ emissions in the eastern and central regions, in the western region there is an obvious upward trend that seems to be much more pronounced when spatial dependence is considered. This means that the western cities will experience a persistent and harsh increase of carbon dioxide emissions if there are no specific intervention policies in place. Kang et al. [19], He and Lin [4], Yin et al. [3], and many others have provided similar evidence.

### 4.3. Robustness Check

In order to evaluate the statistical sensitivity of our findings in regard to other pollutant indicators, we introduced the emissions intensity levels from industrial powder/dust (Dust), chemical oxygen demands (COD), and industrial wastewater (Water) into Equation (5). All three of these pollutants are included in the 11th and 12th Five-Year Plans for Environmental Protection and are set by the central authority as obligatory indicators to be used in the assessment and promotion of local officials in China. The results for the three pollutants consistently documented the validity of the EKC hypothesis in China. In addition, to identify and eliminate any possible endogenous problems, we introduced the lagged real per capita GDP as a proxy for economic growth. All of the empirical results are consistent with our findings as presented in the previous section. The results of our robustness checks are not reported here due to the limitation of article length, but they are available on request.

## 5. Discussion

A theoretical argument persists in the relevant literature regarding the central-and-local government relationships in environmental governance, ranging from advocates of a decentralized system to utilize the local governments’ operational flexibility, to a proposal for a centralized (or recentralized) system in order to overcome the “selective implementation” of national policies at local levels [9,23]. Our findings indicate that the ecological transformation of the IGC can achieve a kind of balance between centralization and decentralization, by combining the environmental TRS and the official ranking tournament. In this way, the incorporation of top-down bureaucratic control and local autonomy not only inspires local officials’ adaptability and activity in reducing pollutant emissions, but it also strengthens the officials’ responsiveness to performance rankings. Peer pressure caused by performance rankings can accelerate the enforcement of national environmental policy [8,31].

A set of more detailed and forceful ecological assessment systems for hierarchical promotions should be developed and instituted in the national efforts to reduce the CO_2_ emissions and other ecological damages. First, a supervisory system for the process of environmental governance should be developed and included in the cadre assessment system, and it should include public environmental information disclosures, response actions to citizens’ multiple requests for ecological products, and lifelong accountability investigations into the ecological damage caused by governors’ administrative decisions. Second, in light of the critical role of information in the processes of policy formulation and transfer, indicator assignment and measurement, and official performance identification and assessment, it is quite essential to establish a standard and unitary platform for improving information transparency and efficiency. Third, our estimation results indicate that the intergovernmental competition regarding environmental performance will inherently handicap the technology distribution across the competing local governments. Therefore, to encourage localities to cooperate with their neighboring governments and achieve an overall improvement in the entire regional environment, the central authority should develop an integrated assessment system of environmental performance that is conducted on the basis of city agglomerations or economic zones, rather than just on each single local government and its principal leader.

We also found that the positive environmental impacts of IGC exist primarily among cities with similar economic levels, rather than in those with apparent economic disparities. Therefore, it is quite essential to accommodate the discrepancies in regulatory effects by forming and implementing national environmental regulations [3,19]. For example, in less-developed regions (e.g., the western cities), measures should be taken to counterbalance the constant-growth obstacle to environmental preservation, and to reduce the economic costs of pollution abatement, through such actions as increasing environment-based special transfer payments by the central government to underdeveloped provinces, functionally improving a “green tax” for balancing the environmental cost-benefit ratios across provinces, and developing and distributing technologies for cleaner production and more effective energy use.

This research is subject to a few limitations, such as the constraints on available data capturing the distinct geomorphological and meteorological characteristics of the eastern, central, and western regions, which limit further analyses and make this study just a snapshot of the pollution–income relationship in different regions. Furthermore, we acknowledge that other factors not included in our estimation model could also affect the EKC forms, such as the public awareness, energy structure, and price levels for low- and high-polluting fuels. In particular, an explanatory variable of the individual characteristic of local officials (e.g., age and political ties) should be considered in the implementing analysis of top-down environmental policies, as this would enable understanding of how governors use environmental performance as a tactic in peer IGC activities. We leave this work for the future.

## 6. Conclusions

This research investigates the degree to which the ecological transformation of intergovernmental competition that began in 2007 changes the form of the environmental Kuznets curve, and how that relationship varies for different pollutant indicators (i.e., SO_2_ and CO_2_) and different regional divisions (i.e., the eastern, central, and western regions of China). We do so by constructing a dataset from 274 cities in mainland China for the period 2007 to 2017 and applying an SDM model that was based on a multivariate conceptual framework and combined the STIRPAT specification. Our estimation results demonstrate a consistently inverted U-shaped relationship between income and SO_2_ emissions in all three regions, whereas with CO_2_ emissions as the pollution indicator, the EKC hypothesis holds only in eastern and central cities, and a positive linear income–CO_2_ nexus is found in the western region. Spatial analysis reveals that whereas the IGC flattened the inverted U-shaped curves between income and SO_2_ emissions, it also led to a higher economic cost corresponding to the turning point of the EKC for CO_2_ emissions. The findings indicate that the ecological transformation of the IGC has facilitated a positive up–down yardstick competition in the strategic interactions of sustainable development across local Chinese governments, which can lead to a kind of balance between centralization and decentralization by inspiring local officials’ adaptability and activity in reducing pollutant emissions and strengthening the officials’ responsiveness to performance rankings.

## Figures and Tables

**Figure 1 ijerph-19-14989-f001:**
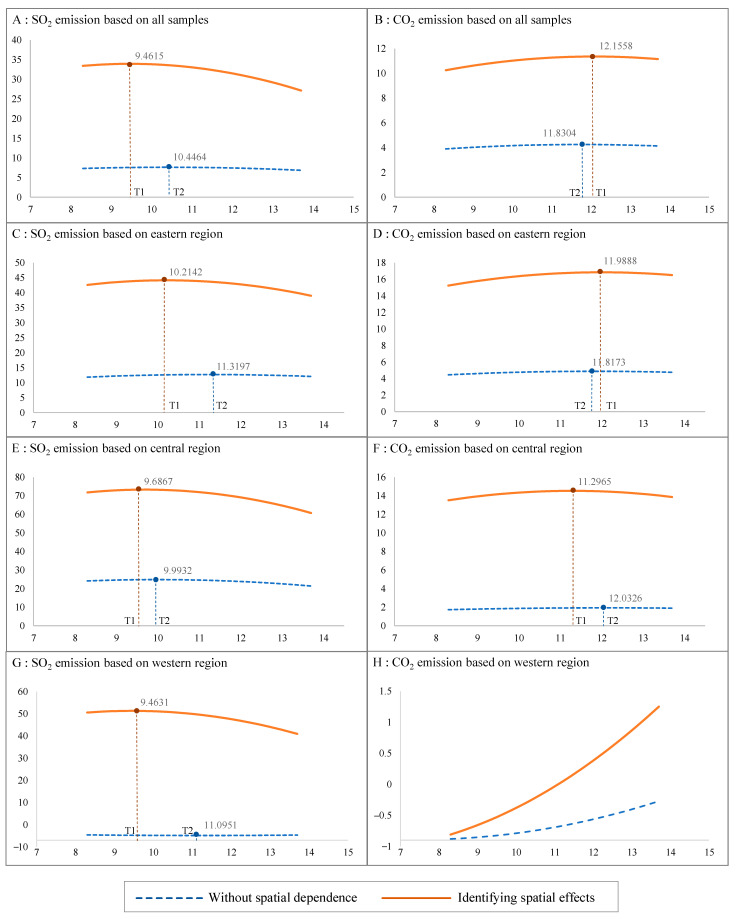
The EKC and its turning points with and without identifying spatial effects, based on different regional divisions.

**Table 1 ijerph-19-14989-t001:** Review of the EKC literature that takes into account the effects of spatial dependence.

Author	Emissions (Turning Point in Yuan) and EKC Form	Regions /Periods	Spatial Econometric Model (Spatial Weights Matrix)	Identification of Spatial Effects
Hao et al. [2]	Per capita coal consumption (39,692–48,521): Inverted-U	29 provinces, 1995–2012	SDM (distance between each two capital cities)	Leading to higher turning point
Zhao et al. [12]	SO_2_ emissions (4057, 24,484): Inverted-N	30 provinces, 1999–2017	SDM (geographically adjacent or not)	Leading to higher turning point
Hao and Peng [5]	Per capita energy consumption: Inverted-U	30 provinces, 1994–2014	SDAR (geographically adjacent or not)	Leading to higher turning point
Kang et al. [19]	CO_2_ emissions (1480.30, 82,677.27): Inverted-N	30 provinces, 1997–2012	SDM (geographically adjacent or not)	Leading to higher turning point
Hao et al. [10]	Self-assessed environmental index (9500): inverted-U	30 provinces, 2006–1015	SDM (geographically adjacent or not)	Leading to lower turning point
Jiang et al. [11]	SO_2_ emission (already passed): downward trend	30 provinces, 2005–2017	SAR (geographically adjacent or not)	Proven to exist
Han et al. [21]	Nitrogen Dioxide emission: positive linear	333 cities, 2016–2018	SAR (geographically adjacent or not)	Proven to exist
Xie et al. [22]	PM2.5 concentrations (25,336.47): inverted-U	249 cities, 2015	SAR (geographically adjacent or not)	Proven to exist

Notes: SDM = spatial Durbin model; SDAR = spatial dynamic lag model; SAR = spatial lag model.

**Table 2 ijerph-19-14989-t002:** Definitions and descriptive statistics for the variables.

Variable	Definition	Mean	S.D.	Max.	Min.
lnSO_2_	Logarithm of the amount of sulfur dioxide emissions (Tons).	10.410	1.112	13.115	0.693
lnCO_2_	Logarithm of the amount of carbon dioxide emissions (thousand Tons).	16.846	0.737	18.68	14.553
ln*PGDP*	Logarithm of per capita real GDP (2007 Yuan).	10.417	0.657	13.056	8.377
ln*PD*	Logarithm of people per square kilometer (people/km^2^).	5.751	0.897	7.882	1.574
ln*RIE*	Logarithm of the index of RIE developed by the PKUCER.	3.749	0.769	4.605	0.024
ln*GOV*	Logarithm of the ratio of municipal fiscal expenditure to GDP (%).	2.749	0.437	5.427	0.324

**Table 3 ijerph-19-14989-t003:** Test results of Moran’s *I* for global spatial dependence (2007–2017).

Years	Case 1: ln*PE* = lnSO_2_			Case 2: ln*PE* = lnCO_2_		
	*Wpro*	*Weco*	*Wpe*	*Wpro*	*Weco*	*Wpe*
2007	0.340 ***	0.154 ***	0.402 ***	0.404 ***	0.409 ***	0.625 ***
2008	0.398 ***	0.133 ***	0.449 ***	0.408 ***	0.409 ***	0.626 ***
2009	0.410 ***	0.133 ***	0.455 ***	0.397 ***	0.410 ***	0.618 ***
2010	0.456 ***	0.125 ***	0.504 ***	0.392 ***	0.412 ***	0.615 ***
2011	0.345 ***	0.139 ***	0.409 ***	0.377 ***	0.417 ***	0.607 ***
2012	0.369 ***	0.137 ***	0.433 ***	0.375 ***	0.418 ***	0.605 ***
2013	0.548 ***	0.033 *	0.601 ***	0.346 ***	0.413 ***	0.587 ***
2014	0.398 ***	0.071 ***	0.448 ***	0.340 ***	0.416 ***	0.584 ***
2015	0.368 ***	0.113 ***	0.431 ***	0.349 ***	0.417 ***	0.592 ***
2016	0.354 ***	0.080 ***	0.411 ***	0.345 ***	0.420 ***	0.589 ***
2017	0.295 ***	0.098 ***	0.351 ***	0.329 ***	0.426 ***	0.579 ***

Note: *** *p* < 0.01, * *p* < 0.1.

**Table 4 ijerph-19-14989-t004:** Estimation results of two-way FE models without and with identifying spatial effects.

Variables	Two-Way FE Models without Spatial Dependence		Two-Way FE Spatial Durbin Models	
	Case 1: ln*PE* = lnSO_2_	Case 2: ln*PE* = lnCO_2_	Case 1: ln*PE* = lnSO_2_	Case 2: ln*PE* = lnCO_2_
*Wpe* × ln*PE*			0.465 *** (7.86)	0.759 *** (37.02)
ln*PGDP*	1.473 *** (3.23)	0.726 *** (11.12)	1.711 * (1.91)	0.524 *** (3.80)
ln*PGDP*^2^	−0.071 *** (−3.21)	−0.031 *** (−9.77)	−0.079 ** (−1.86)	−0.020 *** (−3.16)
ln*PD*	−0.350 (−1.41)	0.307 *** (8.64)	−0.109 *** (−0.26)	0.053 * (0.93)
ln*RIE*	0.011 (0.87)	0.002 (1.17)	0.013 (1.03)	0.001 (0.22)
ln*GOV*	0.011 (0.19)	−0.001 (−0.09)	−0.038 (−0.48)	0.020 (1.45)
*Wpe* × ln*PGDP*			2.075 * (1.73)	−0.076 (−0.44)
*Wpe* × ln*PGDP*^2^			−0.122 ** (−2.10)	0.002 (0.27)
*Wpe* × ln*PD*			−0.837 * (−1.77)	0.126 (1.50)
*Wpe* × ln*RIE*			0.018 (0.60)	0.007 ** (2.53)
*Wpe* × ln*GOV*			−0.291 *** (−2.61)	−0.009 (−0.54)
*(Adj.) R* ^2^	0.807	0.991	0.529	0.769
Observations	3014	3014	3014	3014
LM test (H0: pooled OLS)	6473.89 ***	14113.06 ***		
LR test (H0: one-way FE)	1050.32 ***	1369.39 ***	469.66 ***	424.05 ***
Hausman test (H0: random effects)	85.51 ***	876.48 ***	10.93 **	196.63 ***

Note: *** *p* < 0.01, ** *p* <0.05, * *p* < 0.1. *t*-statistics are reported in parentheses.

**Table 5 ijerph-19-14989-t005:** Diagnostic tests of spatial specification.

Determinants	Two-Way FE Models without Spatial Dependence		Determinants	Two-Way FE Spatial Durbin Models	
	Case 1: ln*PE* = lnSO_2_	Case 2: ln*PE* = lnCO_2_		Case 1: ln*PE* = lnSO_2_	Case 2: ln*PE* = lnCO_2_
LM spatial lag	886.95 ***	420.26 ***	Wald test spatial lag	20.97 ***	9.60 **
Robust LM spatial lag	5.793 **	21.901 ***	LR test spatial lag	1282.54 ***	2419.11 ***
LM spatial error	1549.6 ***	1469.7 ***	Wald test spatial error	41.63 ***	81.37 ***
Robust LM spatial error	668.45 ***	1071.3 ***	LR test spatial error	113.71 ***	435.30 ***

Note: *** *p* < 0.01, ** *p* <0.05.

**Table 6 ijerph-19-14989-t006:** Estimation results of two-way FE SDM models based on regional divisions.

Variables	Case 1: ln*PE* = lnSO_2_			Case 2: ln*PE* = lnCO_2_		
	Eastern	Central	Western	Eastern	Central	Western
*Wpe* × ln*PE*	0.345 *** (2.71)	0.552 *** (13.57)	0.466 *** (10.13)	0.813 *** (34.39)	0.827 *** (16.04)	0.637 *** (15.08)
ln*PGDP*	3.175 ** (1.407)	5.976 *** (4.16)	−2.053 (−1.32)	0.511 *** (4.25)	0.107 (0.76)	0.609 ** (1.96)
ln*PGDP*^2^	−0.141 ** (−2.19)	−0.291 *** (−3.81)	0.094 (1.23)	−0.021 *** (−4.05)	−0.003 * (−0.88)	−0.018 (−1.20)
ln*PD*	−1.531 ** (−2.09)	0.062 (0.15)	0.732 (1.10)	−0.041 (−0.56)	0.135 *** (3.06)	0.138 (1.24)
ln*RIE*	−0.005 (−0.37)	0.031 (1.50)	0.034 (1.02)	0.002 * (1.77)	−0.002 (−1.58)	−0.001 (−0.24)
ln*GOV*	−0.037 (−0.45)	0.199 (0.95)	−0.123 (−1.00)	0.006 (0.76)	0.013 (1.06)	0.055 (1.47)
*Wpe* × ln*PGDP*	2.281 * (1.10)	0.604 (0.25)	7.839 *** (3.83)	0.015 (0.08)	0.335 (1.26)	−0.788 * (−1.84)
*Wpe* × ln*PGDP*^2^	−0.129 * (−1.69)	−0.049 (−0.41)	−0.401 *** (−3.88)	−0.001 (−0.04)	−0.017 (−1.29)	0.031 (1.27)
*Wpe* × ln*PD*	−0.342 (−0.39)	0.955 (1.16)	0.944 (1.39)	0.009 (0.08)	0.046 * (0.55)	0.357 ** (2.55)
*Wpe* × ln*RIE*	0.045 (1.26)	−0.051 (−1.37)	0.047 (1.05)	−0.003 (−1.19)	−0.007 ** (2.57)	−0.019 ** (2.49)
*Wpe* × ln*GOV*	−0.42 *** (−3.43)	−0.346 (−1.69)	−0.272 (−1.07)	−0.012 (−0.93)	0.019 (1.06)	0.008 (0.15)
*R^2^*	0.265	0.274	0.271	0.731	0.803	0.822
Observations	1078	1100	836	1078	1100	836

Note: *** *p* < 0.01, ** *p* <0.05, * *p* < 0.1. *t*-statistics are reported in parentheses.

**Table 7 ijerph-19-14989-t007:** Direct, indirect, and total marginal effects of SDMs and their EKC turning points, compared with those without identifying spatial effects.

Variables	Case 1: ln*PE* = lnSO_2_				Case 2: ln*PE* = lnCO_2_			
	All Samples	Eastern	Central	Western	All Samples	Eastern	Central	Western
Panel A. Tests on EKC specifications under the SDM models								
*A*1 *Direct effects:*								
ln*PGDP*	2.205 ** (2.52)	3.168 *** (2.69)	6.903 *** (4.12)	−0.854 (−0.53)	0.672 *** (4.21)	0.752 *** (5.67)	0.372 ** (2.07)	0.513 * (1.32)
ln*PGDP*^2^	−0.106 *** (−2.55)	−0.164 *** (−2.66)	−0.341 *** (−4.01)	0.032 (0.40)	−0.028 *** (−3.49)	−0.032 *** (−5.34)	−0.015 * (−1.67)	−0.013 (−0.65)
Turning point	10.4341	11.0533	10.1296		12.0163	11.9034	12.7172	19.7719
*A*2 *Indirect effects:*								
ln*PGDP*	4.968 *** (2.89)	5.037 * (1.73)	8.229 * (1.72)	11.709 *** (3.81)	1.212 ** (2.44)	2.057 *** (3.00)	2.202 ** (2.16)	−0.901 (−0.85)
ln*PGDP*^2^	−0.273 *** (−3.27)	−0.260 ** (−1.89)	−0.440 ** (−1.89)	−0.606 *** (−3.96)	−0.050 ** (−2.05)	−0.086 *** (−2.65)	−0.099 ** (−1.98)	0.048 (0.92)
Turning point	9.0856	9.6861	9.3441	9.6681	12.1646	12.0204	11.0874	
*A*3 *Total effects:*								
ln*PGDP*	7.174 *** (3.94)	8.655 *** (4.13)	15.132 *** (2.73)	10.855 *** (2.88)	1.883 *** (3.19)	2.809 *** (3.75)	2.574 ** (2.26)	−0.388 (−0.29)
ln*PGDP*^2^	−0.379 *** (−4.27)	−0.424 *** (−4.32)	−0.78 1*** (−2.88)	−0.574 *** (−3.07)	−0.078 *** (−2.64)	−0.117 *** (−2.65)	−0.114 ** (−2.05)	0.035 (0.53)
Turning point	9.4615	10.2142	9.6867	9.4631	12.1558	11.9888	11.2965	
Panel B. Tests on EKC specifications without considering spatial dependence								
ln*PGDP*	1.473 ***(3.23)	2.264 ** (2.51)	4.972 *** (5.27)	−0.877 (−1.09)	0.726 *** (11.12)	0.827 *** (7.07)	0.316 *** (2.74)	−0.239 ** (−1.99)
ln*PGDP*^2^	−0.071 *** (−3.21)	−0.101 ** (−2.39)	−0.249 *** (−5.34)	0.039 (0.97)	−0.031 *** (−9.77)	−0.035 *** (−6.45)	−0.013 ** (−2.31)	0.016 ** (2.56)
Turning point	10.4464	11.3197	9.9932	11.0951	11.8304	11.8173	12.0326	7.7283

Note: *** *p* < 0.01, ** *p* <0.05, * *p* < 0.1. *t*-statistics are reported in parentheses. All regressions are conducted based on two-way FE SDMs. Only the coefficients of ln*PGDP* and ln*PGDP*^2^ are reported to save space.

## Data Availability

All data and material generated or used during the study appear in the submitted article.

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
