# Peer review of "Reexamining the Environmental Kuznets Curve in Chinese Cities: Does Intergovernmental Competition Matter?"

_ijerph, 2022, doi:10.3390/ijerph192214989_

Round 1

Reviewer 2 Report

This is an excellent work with interesting and rigorous analysis and important discussion. The conclusions that the ecological transformation of the IGC can  balance  centralization and decentralization is highly important, as well as the options for regional incentives and environmental service fees to less developed areas (e.g. the Western region). These points could be given greater prominence in the Abstract. The findings have potential to build further important analysis into motivations for improved environmental outcomes.

Although the paper includes comment on data robustness check, the paper would benefit from a short limitations section rather than only referring to the work done. Several sentences to summarize the limitations and checks would strengthen the confidence of findings, particularly to note potential other sources of contaminates e.g. burning of agricultural biomass, volcanic activity, etc, also the geomorphology of different regions and other aspects that the authors considered that could affect results other than the EKC hypothesis. 
